# A Particulate Matter Concentration Prediction Model Based on Long Short-Term Memory and an Artificial Neural Network

**DOI:** 10.3390/ijerph18136801

**Published:** 2021-06-24

**Authors:** Junbeom Park, Seongju Chang

**Affiliations:** Department of Civil and Environmental Engineering, Korea Advanced Institute of Science and Technology, Deajeon 34141, Korea; jb.park@kaist.ac.kr

**Keywords:** air pollution, artificial neural network, long short-term memory, fine particulate matter, prediction model

## Abstract

Many countries are concerned about high particulate matter (PM) concentrations caused by rapid industrial development, which can harm both human health and the environment. To manage PM, the prediction of PM concentrations based on historical data is actively being conducted. Existing technologies for predicting PM mostly assess the model performance for the prediction of existing PM concentrations; however, PM must be forecast in advance, before it becomes highly concentrated and causes damage to the citizens living in the affected regions. Thus, it is necessary to conduct research on an index that can illustrate whether the PM concentration will increase or decrease. We developed a model that can predict whether the PM concentration might increase or decrease after a certain time, specifically for PM${}_{2.5}$ (fine PM) generated by anthropogenic volatile organic compounds. An algorithm that can select a model on an hourly basis, based on the long short-term memory (LSTM) and artificial neural network (ANN) models, was developed. The proposed algorithm exhibited a higher F1-score than the LSTM, ANN, or random forest models alone. The model developed in this study could be used to predict future regional PM concentration levels more effectively.

## 1. Introduction

With increased energy consumption due to rapid urbanization and industrialization, vehicle exhaust and other polluting emissions have caused serious damage to human health and the environment [1,2,3,4]. Air pollution is causing various socio-economic problems, and interest in air pollution has been increasing worldwide. The World Health Organization has reported that 92% of the world is influenced by air pollution, and more people are being directly damaged by it every year. Thus, air pollution can affect lives and threaten public health [5,6,7,8]. Particles are small substances in the air, and extremely small particles with a diameter of less than 10 micrometers are called particulate matter (PM). PM refers to small dust particles that float in the air and are not visible to the human eye. As PM is minuscule, it can infiltrate the body and adversely affect human health. PM includes foreign substances, such as metals, nitrates, and sulfates, that can cause respiratory and other types of diseases. PM emissions can be classified into those originating from natural sources, such as forest fires and yellow dust, and artificial sources, such as fumes and factories. Naturally occurring PM is classified as biogenic volatile organic compounds (BVOCs), while artificially occurring PM is classified as anthropogenic volatile organic compounds (AVOCs). At present, predicting the air pollutant concentration is considered an effective method for managing air quality [9].

Over the past several decades, most researchers have analyzed and predicted the concentrations of air pollutants, in order to reduce the damage of air pollution and enhance prediction accuracy [10,11,12,13,14]. A study was conducted to predict the number of hospital admissions for patients with respiratory diseases, based on artificial neural network (ANN)-mediated fine dust (PM${}_{10}$) exposure estimation. Average daily temperature, relative humidity, and PM${}_{10}$ concentration datasets were collected for three years (2007–2009); as a result, the developed model was found to be relevant for analyzing potential health risks [15]. As for ensemble methods, which combine multiple learning algorithms to obtain better predictive performance, the development of an ensemble multistep-ahead framework and dynamic ensemble mechanisms has also been suggested. In addition, a model considering various variables was proposed by combining ensemble learning and multi-layer perceptron (MLP) [16,17,18,19,20]. Many diverse studies have been conducted, in which particle swarms have been optimized using swarm intelligence in combination with ANNs [21,22,23,24,25]. In the field of residual modeling, multiple studies have been conducted on hybrid models, in order to improve the performance of predicting PM concentration levels [26,27]. Particularly in Asia, as the damage caused by air pollutants increases, PM prediction is becoming an important issue to be addressed. An air quality prediction model has been proposed based on the grey accumulated generating technique and Holt–Winter method. Data collected from 2014 to 2016 were used in the study, and seasonal characteristics were reflected in the algorithm to enhance its accuracy [28]. Other research has been conducted using artificial neural network (ANN) and K-nearest neighbor methods to predict PM${}_{10}$ concentration, where the prediction accuracy was improved by obtaining the optimal hyperparameters [29]. The predictive performance of a machine learning model could have different effects, depending on the composition of the dataset used. The prediction performance for PM${}_{10}$ concentration prediction performance enhancement was attempted, using a deep neural network, support vector machine, and random forest model, and the authors analyzed the environmental factors influencing PM concentration levels [30].

Among deep learning models, long short-term memory (LSTM) and the denoising autoencoder (DAE) can be helpful in prediction, by remembering previous values [31]. A prediction algorithm, based on a geographically and temporally weighted regression model (which improved the model accuracy), was introduced to analyze PM influencing city-scale fog pollution [32]. In another study, LSTM-based transfer learning with a limited Chinese air quality data sample showed improved PM prediction performance [33]. They improved the learning efficiency and prediction accuracy by using high-performance screening features for transfer learning. A method for improving prediction performance by solving the PM data outlier problem has also been proposed. They demonstrated that PM prediction performance can be increased by detecting outliers in PM data and applying a heuristic correction algorithm [34]. Studies to predict agricultural carbon dioxide (CO${}_{2}$) have also been conducted. A model has been introduced to predict the CO${}_{2}$ generated in the agricultural sector in Iran [35]. They generated the features of an inclusive multiple model based on multiple regression, Gaussian process regression, and ANN models, which improved the CO${}_{2}$ prediction accuracy.

Studies have also been conducted to improve the prediction performance using convolutional neural network (CNN) algorithms. For instance, factor separation based on empirical mode decomposition improved model prediction performance when adopting a hybrid model for PM${}_{2.5}$ prediction, which was trained based on a CNN [36]. In one study, researchers collected and analyzed PM${}_{2.5}$ data over a three-year period (2015–2017) in Beijing, China [37]. They analyzed the PM data characteristics with a CNN and a conducted time series data analysis using LSTM. In addition, they developed a CNN–LSTM model for improved PM${}_{2.5}$ prediction performance and tested the model performance at 384 stations in Beijing. Another study, based on the use of a CNN, showed the development of an extended convolutional long short-term memory neural network model, which improved PM prediction accuracy at 1233 stations in Beijing [38].

However, PM must be forecasted before it increases to a high concentration and, consequently, causes damage to citizens in the affected area. Thus, creating an index indicating the probability that the concentration of PM will increase or decrease is crucial. We developed a concentration prediction model for fine PM (PM${}_{2.5}$), which has been classified as an AVOC. This model can predict whether the PM concentration will increase or decrease at a certain time, based on LSTM and artificial neural network (ANN) models; the model performance was evaluated according to its F1-score.

## 2. Data Collection and Processing Methods

### 2.1. PM Measurement Site

To analyze PM, we utilized data collected in Cheongnyang-ri, Dongdaemun-gu (Location A), and Yangjae 1-dong (Location B) in Seocho-gu, Seoul, South Korea (see Figure 1). To verify the accuracy of the proposed algorithm, the datasets were analyzed and the model was evaluated, based on the two locations displaying differences in temperature and carbon content near the Han River. Location A is approximately 6.3 km above the Han River, while Location B is roughly 6.2 km below the Han River.

Reference mean temperatures in 2020 for Location A were 5.61 °C for January–March, 19.03 °C for April–June, 25.06 °C for July–September, and 15.78 °C for October–December. For Location B, the reference mean temperatures were 4.65 °C for January–March, 17.39 °C for April–June, 23.28 °C for July–September, and 13.17 °C for October–December. Thus, the average temperature of Location A was 1.7 °C higher than that of Location B. The monthly industrial carbon concentration of Location B was approximately 1.03-times higher than that of Location A.

### 2.2. Meteorological and PM Datasets

Table 1 describes the dataset fields used in this study. The data were measured every 10 min, including temperature, humidity, wind direction, wind speed, AVOC PM concentration, BVOC PM concentration, and total carbon concentration. The number of data collected was 51,608 for Location A and 48,977 for Location B. As these datasets were obtained in 10 min increments, hourly average values were calculated and used for the necessary data analyses.

Figure 2 displays the average meteorological data for Locations A and B. Meteorological data for the research period obtained from the website of the Korea Forest Service website were used for this research. The datasets included hourly values of temperature, humidity, wind direction, and wind speed. Meteorological data were referenced for the analysis due to the influence of the weather conditions on PM concentration. As shown in Figure 3, there was a decreasing trend in PM concentration from the 150th to the 250th day, which was approximately from May to August. The overall trend of the PM concentrations was very irregular. The potential reason for this PM concentration irregularity is that it would have been influenced by neighboring countries and industrial facilities around the measurement site.

### 2.3. PM Prediction Model Structure

This research attempted to study whether predicted PM concentration values would increase or decrease after 3 and 5 h, when using the LSTM and ANN models to approximate the AVOC PM${}_{2.5}$ concentration. The model analysis and prediction process in this study are illustrated in Figure 4.

### 2.4. Pre-Processing

#### 2.4.1. Principal Component Analysis

Figure 5 displays the correlations among all measured features. A correlation coefficient closer to −1.0 represents a strong negative correlation, while a correlation coefficient closer to +1.0 represents a strong positive correlation. Consequently, a feature having a correlation close to −1.0 or +1.0 is considered to be significant. Figure 5 illustrates that the temperature exhibited a negative correlation with AVOC and BVOC PM. In addition, humidity exhibited a negative correlation with AVOC PM${}_{10}$.

PCA generates a few new PCs that explain the greatest proportion of the data. The PCs are uncorrelated artificial variables, converted from the original variables by a linear combination. Concerning the order of importance based on the PC variance, the dimensions can be reduced to ensure that the first few PCs explain as much of the total variation (information) as possible in the original data. It was assumed that the PCA had a probability vector $X={\left(x_{1},...,x_{n}\right)}^{T}$, a population mean vector $\mu ={\left(\mu_{1},...,\mu_{n}\right)}^{T}$, and a population covariance matrix $\sum ={\left\{\sigma_{i}{,}_{j}\right\}}_{i,j=1,\ldots ,n}$. This probability vector was placed at the *n*th dimension, and *n* new co-ordinate axes can be formed through the linear combination of the original variables. The *n* new coordinate axes that facilitate the covariance structure analysis while optimally explaining the data variance are the PCs.

#### 2.4.2. Linear Discriminant Analysis

Linear discriminant analysis (LDA) was developed as a supervised learning method to classify new objects into groups, after creating a discriminant function or rule that can distinguish groups based on information about them. It is often used to reduce dimensions to easily classify the characteristic variables of the data, corresponding to unsupervised learning. The objective is to determine a discriminant function that represents and distinguishes the characteristics of several groups and to determine into which group to classify new observed values. The Fisher LDA is a representative discriminant analysis method for distinguishing between two groups. The Fisher LDA contains the following assumptions: Between groups $G_{1}$ and $G_{2}$, $m_{1}$ data are acquired from $G_{1}$ and $m_{2}$ data are obtained from $G_{2}$. Each dataset is composed of *n* characteristic variables, where $X={\left(x_{1},...,x_{n}\right)}^{T}$, and one categorical target variable *T*; the number of categories (groups) of the target variables is assumed to be two. The population mean vector of *X* is $u_{1}$ for data that belong to the first group and $u_{2}$ for data that belong to the second group, where $u_{1}$ and $u_{2}$ are n×1 vectors. The population covariance matrix of *X* is assumed to be ∑ for the data of both groups, where ∑ is an n×n matrix.
(1)$$y=l^{T}X=l_{1}x_{1}+l_{2}x_{2}+\cdots +l_{n}x_{n},$$
(2)$$u_{1}y=l^{T}u_{1},u_{2}y=l^{T}u_{2},$$
(3)$$|u_{1}y-u_{2}y|=|l^{T}\left(u_{1}-u_{2}\right)|.$$

To distinguish between the two groups $G_{1}$ and $G_{2}$, a linear discriminant function, as shown in Equation (Equation 1), which distinguishes the $G_{1}$ and $G_{2}$ that are as far apart as possible, is obtained. An axis (the linear discriminant function $l^{T}X$ of $x_{1},...,x_{n}$), which makes the intergroup variation larger than the intragroup variation as much as possible, is then found. The group mean of $y=l^{T}X$ is expressed as Equation (Equation 2). A larger intergroup mean difference in the linear discriminant function value *y* indicates an improved distinction between the groups. Furthermore, when all the intragroup differences of the linear discriminant function $y=l^{T}X$ are the same, it is easier to distinguish between the groups when the intragroup variation is smaller.

### 2.5. Artificial Neural Networks

An artificial neural network (ANN) is an supervised learning algorithm derived from the nervous system of higher organisms. The basic units of an ANN are the neurons. In this study, MLP was used among the various ANN models. The neurons comprise a neural network, which aggregates multiple input signals into one and outputs the signal through an activation function; then, the produced output enters another neuron as an input. As shown in Equations (4) and (5), the sigmoid function was used as the activation function. The output, to which the weight and bias are added, corresponding to each neuron, is determined as the output of the neuron using the tangent sigmoid function. Here, *N* is the number of input neurons, *i* and *k* are the number of input variables and biases, $x_{i}$ is the input variable, and $w_{ij}$ and $b_{j}$ are the weights and biases of the input variable and the hidden layer, respectively.
(4)$$Net_{j}=\sum_{i=0}^{N-1}w_{ij}x_{i}+b_{j},$$
(5)$$f_{j}\left(Net_{j}\right)=1/\left(1+e^{-Net_{j}}\right).$$

### 2.6. Long Short-Term Memory

LSTM is a recurrent neural network (RNN) architecture that memorizes values in random intervals. Theoretically, SimpleRNN can maintain timestamp data for less than time *t*, but, in actuality, long-term dependence generates training problems. Thus, a deeper layer of the feed-forward natural network is more difficult to train. LSTMs are designed to prevent long-term dependence problems. They can selectively flow data in the cell state by discarding or updating new data at a certain moment, using the forget, input, and output gates. First, the forget gate discards or maintains the previous and current data using the sigmoid function. The input gate determines the value stored in the cell state as the product of the results obtained from the sigmoid function and the activation function for the previous and current data. Then, the results of the forget and input gates are added, and the result is updated in the cell state. Finally, the output gate determines the output value by multiplying the value obtained through the sigmoid function for the previous and current data, by that obtained through the activation function in the updated cell state. Figure 6 illustrates the LSTM structure, and Equations (6)–(11) represent the LSTM equations. Here, $h_{(}t-1)$ indicates the data of the previous hidden state; $x_{t}$ indicates the current input data; *w* is the weight; *b* is the bias; $f_{t}$ is the result of the forget gate; $i_{t}$ and $\tilde{C_{t}}$ are the results of the sigmoid and activation functions for the previous and current data, respectively; $C_{t}$ is the result to be updated in the cell state; $o_{t}$ is the result of the output gate; and $h_{t}$ is the final output value.
(6)$$f_{t}=\sigma \left(w_{f}\cdot \left[h(t-1\right),x_{t}\right]+b_{f}),$$
(7)$$i_{t}=\sigma \left(w_{i}\cdot \left[h(t-1\right),x_{t}\right]+b_{i}),$$
(8)$$\tilde{C_{t}}=tanh\left(w_{c}\cdot \left[h(t-1\right),x_{t}\right]+b_{C}),$$
(9)$$C_{t}=f_{t}*C_{t-1}+i_{t}*\tilde{C_{t}},$$
(10)$$o_{t}=\sigma \left(w_{o}\cdot \left[h_{t-1},x_{t}\right]+b_{o}\right),$$
(11)$$h_{t}=o_{t}*tanh\left(C_{t}\right).$$

### 2.7. Model Selection

The model selection process chooses a better fitting model for each time period, based on the algorithm proposed in this study. This model selection scheme is used for the testing dataset, based on the time map generated for each time zone, through the validation process using the training dataset. The model selection process was carried out as follows.

Apply standard scaler and extract variables by PCA and LDA;Calculate the F1-score for each model. For the ANN and LSTM models, store the hourly F1-score values of the validation data.
(12)$$Value_{n}=F1_{1},\ldots ,F1_{n-1};$$Compare the F1-score by model and generate hourly time maps.
(13)$$ANN_{h}=mean\left(\sum F1_{h}\right)\cdot \left(1-std\left(\sum F1_{h}\right)\right),$$
(14)$$LSTM_{h}=mean\left(\sum F1_{h}\right)\cdot \left(1-std\left(\sum F1_{h}\right)\right),$$
(15)$$Map_{Time(h)}=ANN_{h},LSTM_{h};$$Select a model for each hour. Extract values corresponding to the hour using the linear regression model, and select the model showing the highest value;Obtain the prediction result.

Equations (12)–(15) represent the overall model selection process. $Value_{n}$ is a variable that stores the F1-scores in all time zones across the validation data. $ANN_{h}$ and $LSTM_{h}$ are the variables that store each model’s hourly prediction performance. The $Map_{Time(h)}$ variable organizes $ANN_{h}$ and $LSTM_{h}$ into a table to help model selection. Our proposed model selection process is as follows.

First, a time map variable that stores the model performances over 24 h is generated. Then, the hourly F1-scores (F1) of the LSTM and ANN models are determined. Hourly means and variances are then obtained, using the F1-scores. The mean ($\sum F1_{h}$) and $1-std(\sum F1_{h}$) are multiplied, and the results for the LSTM and ANN models are stored in the time map variable. Then, a linear regression model is applied to the hourly scores, such that the model with the highest score can be found. The high scoring model is stored, according to the time map priority. Model selection and prediction are then performed, applying the algorithm based on the time map.

## 3. Results and Discussion

For data analysis, the datasets measured in Locations A and B in Seocho-gu, Seoul, South Korea, were used. We developed and tested a model that predicts whether the PM concentration would increase or decrease in 3 and 5 h, in terms of AVOC PM${}_{2.5}$. The experiment was conducted from 14:00 to 21:00 (when people are generally active). After calculating the mean values by classifying the training and testing datasets based on k-fold cross-validation, the performance of each model was evaluated by comparing the F1-scores.

### 3.1. Experimental Environment

The obtained data were classified into training and testing datasets, based on the use of five folds (see Figure 7). PM concentration profiles showed a large rate of variation, depending on the season. Therefore, in this investigation, the training data were selected to span more than six months, such that seasonal changes could be reflected. In addition, five folds were applied, in order to test the model for accurate performance verification. Each fold consisted of 7–11 months of training data and one month of testing data. In addition, a one-month data segment was separated from the training dataset, for use as validation data. For the first fold, the training dataset was composed of the data starting from January 2020 until July 2020, while the testing dataset was composed of data from August 2020. For the second fold, the training dataset was composed of data from January 2020 to August 2020, while the testing dataset was composed of data from September 2020. Based on this data segmentation scheme, a total of five folds were created, and the experiments was performed using those training and testing datasets. The outcome of the experiment was the average of the F1 values obtained for each fold. For more accurate performance verification, the above-mentioned process was iteratively performed five times, and the resulting average value was used.

As for the LSTM and ANN models, the model with the best epoch value that did not cause overfitting was stored, using the validation data and the ModelCheckpoint function of TensorFlow. The maximum epoch value was set to 500, and the optimized epoch value for each fold was saved. The optimal batch size was 70 for the ANN and 300 for the LSTM, according to the Adam optimizer. In the RF model, the optimal parameters for *n*-estimators and max depth in each fold were found and saved. The average optimal value was about fifty for *n*-estimators and about four for max depth.

### 3.2. Performance Evaluation Method

To verify the model performance, the recall, precision, and F1-score (see Equations (16)–(18)) were calculated and compared. True positive (TP) indicates the case where the correct answer is predicted to be true; false positive (FP) indicates the case where the false answer is predicted to be true; false negative (FN) indicates the case where the correct answer is predicted to be false. In this study, the case where the PM concentration increases was set to be true, while the case where it decreases was set to be false.
(16)Recall=TP/((TP+FN)),
(17)Precision=TP/((TP+FP)),
(18)F1=2∗((Precision∗Recall)/(Precision+Recall)).

Recall is the number of correctly identified positive cases from all of the actual positive cases. This measure is critical when the cost of false negatives is high. Precision is the measure of correctly identified positive cases in all predicted positive cases. This measure is also useful when the cost of false positives is high. The F1-score is the harmonic mean of the precision and recall, which can act as a performance metric when the data label exhibits an unbalanced structure.

### 3.3. PM Prediction Model Analysis

Figure 8 compares the F1-score performances of the ANN and LSTM for different time slots (14:00–21:00, when people are generally active) from May 2020 to December 2020. Although there were time slots when the LSTM model showed slightly increased performance, depending on the training intensity, in certain time slots, the performance of the ANN was slightly higher than that of the LSTM model. The AVOC PM can be transmitted by the wind or be discharged from vehicles and chemical plants. Therefore, the LSTM model, which exhibited sufficient performance in time series data analysis, and the simple neural network-based ANN model demonstrated different hourly performances.

### 3.4. PM Prediction Performance Evaluation

The proposed model prediction performance in this study was evaluated at Locations A and B. It was predicted whether the AVOC PM${}_{2.5}$ would increase or decrease in 3 and 5 h, and the model performances were compared, based on the recall, precision, and F1-score. Figure 9 and Table 2, Table 3, Table 4, Table 5 and Table 6 display the predictions of the PM${}_{2.5}$ concentration at Locations A and B 3 h and 5 h in advance.

To check the performance of the proposed model, the same test process was repeated five times, in order to calculate the average recall, precision, and F1-score. The test results showed that the proposed model had a 1–3% higher F1-score than the LSTM, ANN, and RF models. As the proposed approach selects an effective model on an hourly basis, it showed an improvement in F1-score over LSTM, ANN, and RF; however, the proposed model had a limitation, in that it required the tuning of hyperparameters for each model. Furthermore, in order to effectively apply the proposed model for PM concentration prediction, each model had to be adjusted to maintain a similar performance level.

## 4. Conclusions

Many countries have suffered from various damages due to PM. PM must be forecasted in advance in the affected regions, in order to allow citizens to take necessary precautions. In this study, we investigated a probabilistic index concerning whether the PM value would increase or decrease at certain times. A model for predicting AVOC PM${}_{2.5}$ concentration levels 3 and 5 h in advance was developed by utilizing datasets measured over one year at Locations A and B in Seoul, South Korea. Model testing and validation were performed using five-fold cross-validation. An algorithm that could select the best-performing hourly model, considering LSTM and ANN models, was developed. Our experimental results demonstrated that the model exhibited a 1–3% higher F1-score than the LSTM, ANN, and random forest models.

The most distinctive advantage of our proposed model lies in the fact that the PM concentration prediction performance was improved by selecting the better-performing model between ANN and LSTM, depending on different times. Our suggested approach is considered to be highly scalable, as it could be applied to other PM concentration level prediction tasks, despite the fact that the time-dependent model selection process proposed in this study takes a longer time and requires more resources, as it has to learn and predict with dual models.

As for the future study, further optimization of the model selection proposed in this study should be carried out. This optimization process could harness ensemble techniques and the convergence of swarm intelligence, as well as residual modeling for better PM concentration prediction capability across different global regions.

The following abbreviations are used in this manuscript:

## Figures and Tables

**Figure 1 ijerph-18-06801-f001:**
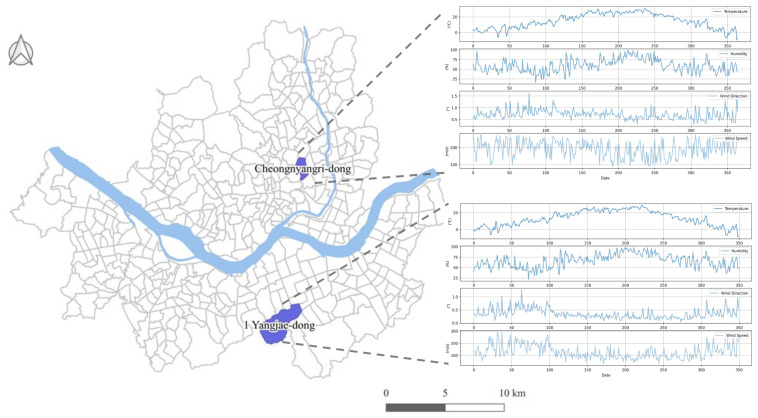
Location map of the study areas.

**Figure 2 ijerph-18-06801-f002:**
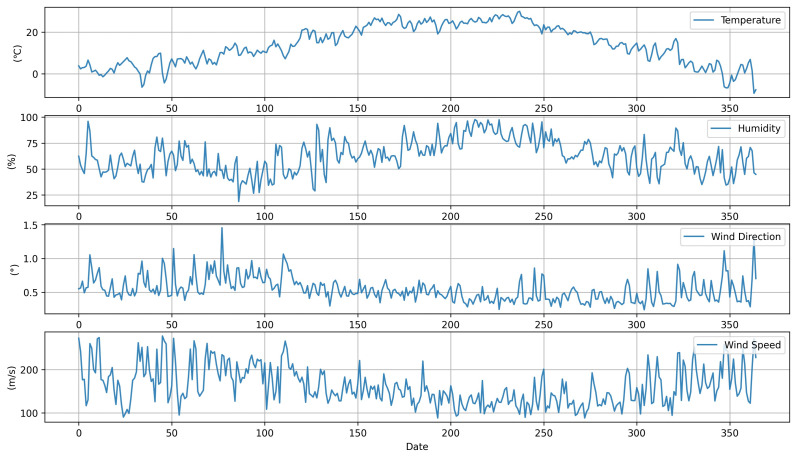
Average meteorological data of Locations A and B.

**Figure 3 ijerph-18-06801-f003:**
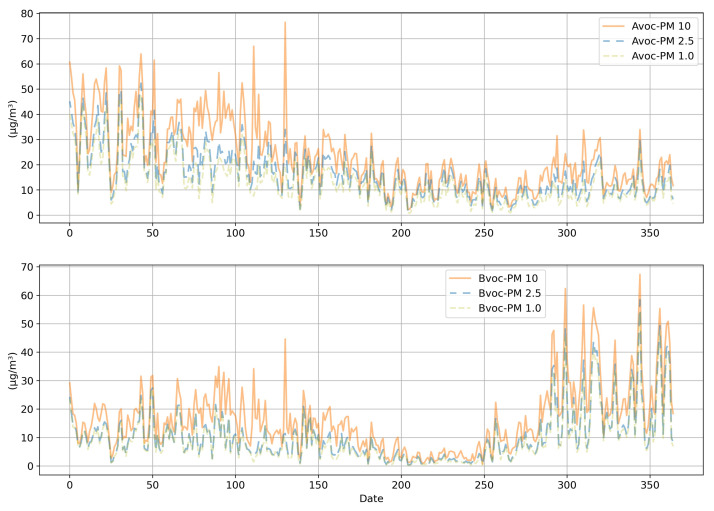
Average PM concentration data of Locations A and B.

**Figure 4 ijerph-18-06801-f004:**
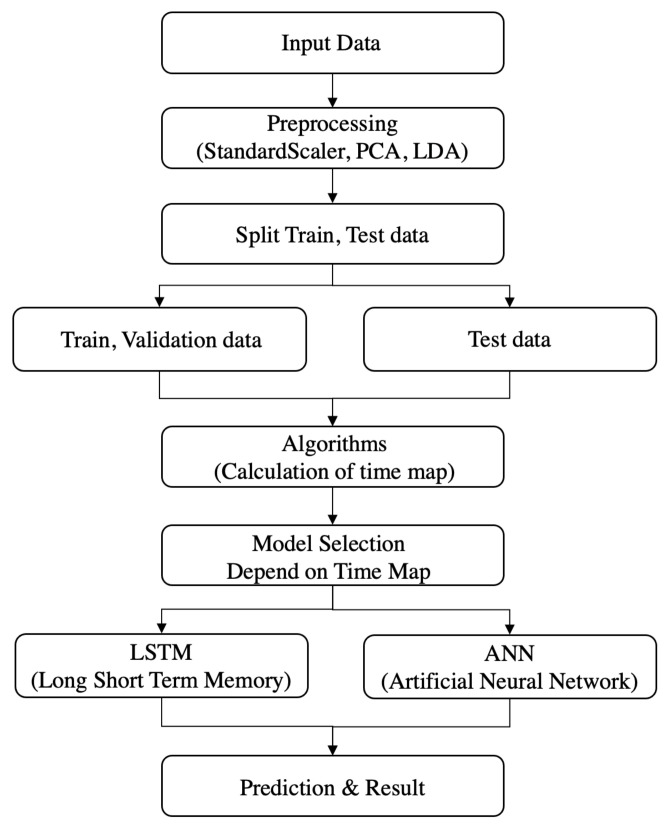
Structure and process of the PM${}_{2.5}$ concentration prediction model.

**Figure 5 ijerph-18-06801-f005:**
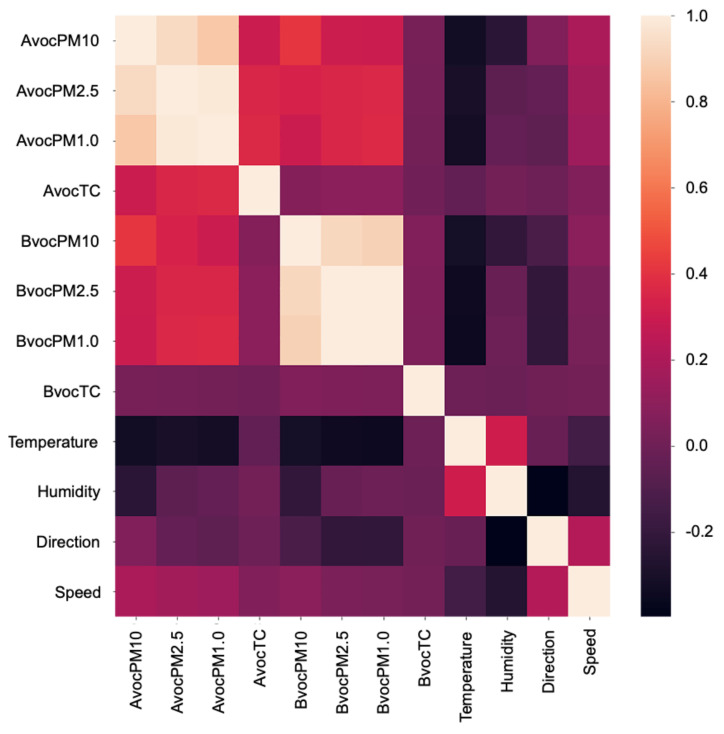
Feature correlation analysis.

**Figure 6 ijerph-18-06801-f006:**
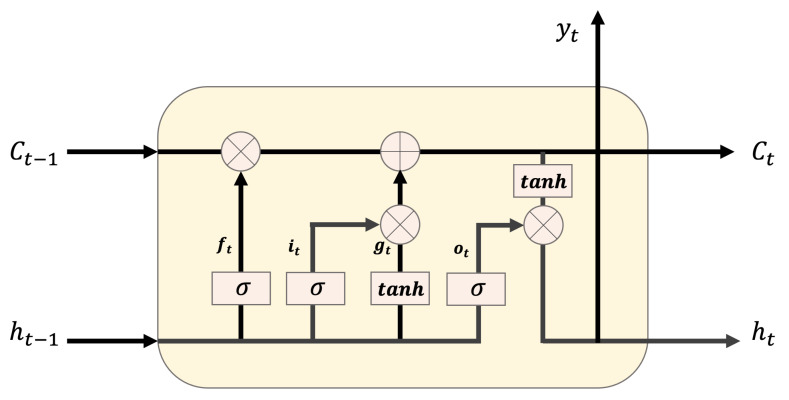
LSTM structure.

**Figure 7 ijerph-18-06801-f007:**
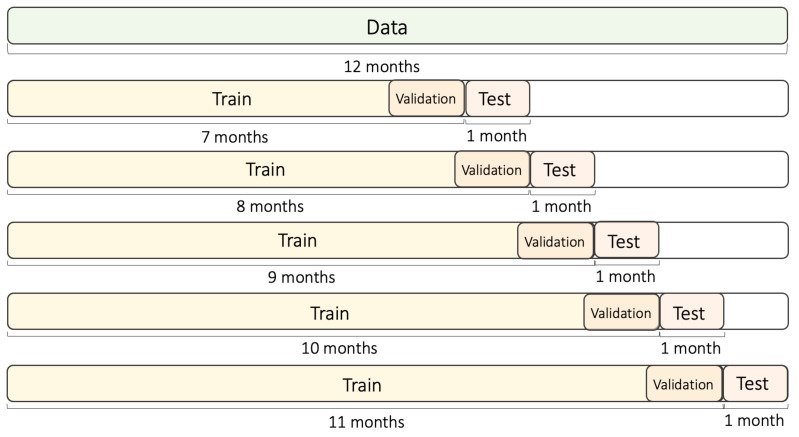
Time series split of the training and testing datasets.

**Figure 8 ijerph-18-06801-f008:**
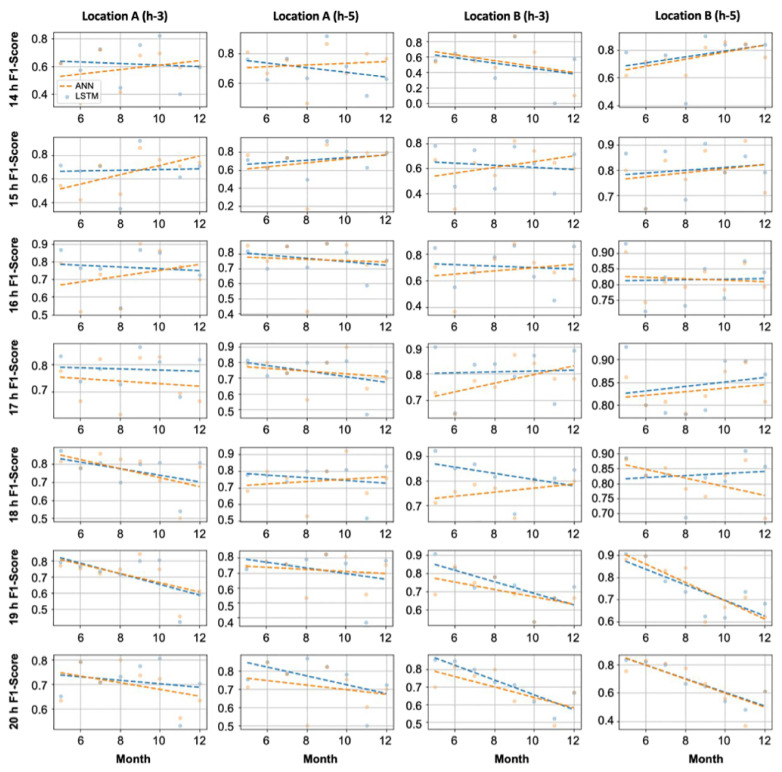
Hourly F1-score performance analysis of the LSTM and ANN.

**Figure 9 ijerph-18-06801-f009:**
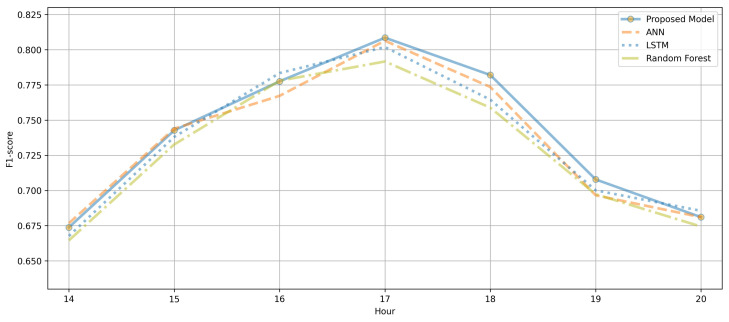
Average of F1-scores for PM${}_{2.5}$ 3 and 5 h predictions at Locations A and B.

**Table 1 ijerph-18-06801-t001:** Variables and descriptions in the datasets.

Variable	Description
Observed time	Year, month, day, hour, and minutes
AVOC-PM${}_{10}$	PM${}_{10}$ value of AVOCs (µg/m^3^)
AVOC-PM${}_{2.5}$	PM${}_{2.5}$ value of AVOCs (µg/m^3^)
AVOC-PM${}_{1.0}$	PM${}_{1.0}$ value of AVOCs (µg/m^3^)
AVOC-TC	Total count of particles related to AVOCs
BVOC-PM${}_{10}$	PM${}_{10}$ value of BVOCs (µg/m^3^)
BVOC-PM${}_{2.5}$	PM${}_{2.5}$ value of BVOCs (µg/m^3^)
BVOC-PM${}_{1.0}$	PM${}_{1.0}$ value of BVOCs (µg/m^3^)
BVOC-TC	Total count of particles related to BVOCs
Temperature	Temperature (°C)
Humidity	Humidity (%)
Direction	Wind direction (°)
Speed	Wind speed (m/s)

**Table 2 ijerph-18-06801-t002:** Model prediction performances after 3 h at Location A.

Method	LSTM	ANN	RF	Proposed Model
Recall	0.829	0.828	0.767	0.821
Precision	0.647	0.646	0.654	0.654
F1-score	0.714	0.713	0.695	0.720

**Table 3 ijerph-18-06801-t003:** Model prediction performances after 5 h at Location A.

Method	LSTM	ANN	RF	Proposed Model
Recall	0.814	0.882	0.793	0.860
Precision	0.688	0.650	0.689	0.679
F1-score	0.737	0.740	0.728	0.749

**Table 4 ijerph-18-06801-t004:** Model prediction performances after 3 h at Location B.

Method	LSTM	ANN	RF	Proposed Model
Recall	0.821	0.888	0.806	0.857
Precision	0.668	0.636	0.656	0.657
F1-score	0.725	0.727	0.713	0.732

**Table 5 ijerph-18-06801-t005:** Model prediction performances after 5 h at Location B.

Method	LSTM	ANN	RF	Proposed Model
Recall	0.832	0.795	0.776	0.818
Precision	0.646	0.653	0.660	0.657
F1-score	0.716	0.709	0.702	0.720

**Table 6 ijerph-18-06801-t006:** Total average performances of the compared models.

Method	LSTM	ANN	RF	Proposed Model
Recall	0.824	0.848	0.786	0.839
Precision	0.660	0.646	0.665	0.662
F1-score	0.723	0.722	0.710	0.730

## Data Availability

The data used to support this study were obtained from the National Institute of Forest Science of the Korea Forest Service. These data are available with as open-access from the web pages of these departments. Details about and references to the data are provided in Section 2.1.

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
