# Peer review of "A Particulate Matter Concentration Prediction Model Based on Long Short-Term Memory and an Artificial Neural Network"

_ijerph, 2021, doi:10.3390/ijerph18136801_

Round 1
Reviewer 1 Report
The paper Systems Particulate Matter Prediction Model Based on Long Short-Term Memory and Artificial Neural Network Models presents an interesting approach to deal with Particulate Matter Prediction. However, the authors should address some issues:
- The references are not enough to understand the state of the art of the Systems Particulate Matter Prediction using Machine Learning techniques;
- Figure 4 should be divided to show the proposed approach's two phases: training and test.
- Why 5 folds are used? How would the proposed approach perform with a number smaller of the folds? Holdout division could be used?
- What the size of the validation set for each fold?
- How the RF model was evaluated? It is not clear.
- The model selection between ANN and LSTM is performed from performance in the training, validation, or test set?
- The conclusions may be improved. For example, what future works should be addressed? Ensembles could be used [1 - 5]? How? Swarm intelligence or Evolutionary algorithms could be used to improve the performance of the proposed method, such as employed by [6 - 10] Residual modeling [11 - 12] could be performed aiming to improve the used ML performance?
[1] An ensemble multi-step-ahead forecasting system for fine particulate matter in urban areas. IK Ahani, M Salari, A Shadman - Journal of Cleaner Production, 2020.
[2] Dynamic ensemble mechanisms to improve particulate matter forecasting. A Bueno, GP Coelho, JRB Junior - Applied Soft Computing, 2020.
[3] Neural-Based Ensembles for Particulate Matter Forecasting
PSGDM Neto et al. IEEE Access, 2021.
[4] Ensemble and enhanced PM10 concentration forecast model based on stepwise regression and wavelet analysis. Y Chen, R Shi, S Shu, W Gao - Atmospheric Environment, 2013.
[5] Development of a stacked ensemble model for forecasting and analyzing daily average PM2. 5 concentrations in Beijing, China
B Zhai, J Chen. Science of The Total Environment, 2018.
[6] SVM kernel based on particle swarm optimized vector and Bayesian optimized SVM in atmospheric particulate matter forecasting. GN Kouziokas - Applied Soft Computing, 2020
[7] Application of a Hybrid Model Based on Echo State Network and Improved Particle Swarm Optimization in PM2.5 Concentration Forecasting: A Case Study of Beijing, China. X Xu, W Ren. Sustainability, 2019.
[8] Air pollutant concentration forecast based on support vector regression and quantum-behaved particle swarm optimization. X Li, A Luo, J Li, Y Li - Environmental Modeling & Assessment, 2019.
[9] Time-series forecasting of pollutant concentration levels using particle swarm optimization and artificial neural networks. FS Albuquerque Filho et al. - Química Nova, 2013.
[10] An approach to improve the performance of PM forecasters. PSG de Mattos Neto, GDC Cavalcanti, F Madeiro. PloS one, 2015.
[11] A hybrid ARIMA and artificial neural networks model to forecast particulate matter in urban areas: The case of Temuco, Chile
LA Díaz-Robles et al. - Atmospheric Environment, 2008.
[12] Nonlinear combination method of forecasters applied to PM time series. PSG de Mattos Neto et al. - Pattern Recognition Letters, 2017.
Author Response
Rebuttal Letter
Title of the Paper: “A Particulate Matter Concentration Prediction Model Based on Long Short-Term Memory and Artificial Neural Networks”
To the Editor and Reviewers:
Here we composed the rebuttal for our previously submitted paper titled as “A Particulate Matter Concentration Prediction Model Based on Long short-Term Memory and Artificial Neural Networks”. We, authors deeply appreciate the reviewers’ constructive comments as well as the associate editor’s decision on letting us have an opportunity for revisiting unclear or overlooked issues in our paper for revision and resubmission. We carefully revised and updated the paper by strictly following the reviewers’ comments and suggestions. We, therefore, tried to maintain our original motivation and framework for developing the paper while correcting and enhancing the details and ambiguous statements based on the given comments. We again thank the editor and the reviewers and sincerely hope a favourable final decision on the paper’s publication opportunity.
Sincerely yours,
Seongju Chang, Ph.D.
Invited Professor
Department of Civil & Environmental Engineering, KAIST, Korea
T) +82-42-350-3627
F) +82-42-350-3610
M) +82-10-2757-4547
May 15, 2021

Reviewer 2 Report
The manuscript introduces a methodology to predict PM using LSTM and ANNs. The topic is interesting
but it is a necessary hard work before publication in a scientific journal.
Some observations:
0- There are a lot of typos in the document;
1- The first paragraph is too long. Please, split it;
2- Line 28 - "The causes of PM" -> the emissions of PM;
3- Line 28 - PM is not a subject, but its concentration or emission. Please correct;
4- Line 40 - Ann -> ANN. Also, this acronym was not defined yet in the text;
5- "for prediction PM10" - to predict PM10;
6- Some state-of-art investigations in the same field were neglected in the introduction,
such as 10.1109/LA-CCI.2016.7885699;
7- In Table 1 the authors must mention the number of samples for each set;
8- In figure 4 the authors mention 5-fold, but this is related to the validation set;
9- Line 122-125 and so on - All variables must be in italics (like "n"), but vectors and matrices in bold, such as X or mu;
10- Line 128 - never start a sentence with an acronym (LDA);
11- Line 138 - Name the Section 2.5 as Artificial Neural Networks;
12- Line 139 - Do - An artificial neural Network (ANN)...;
13- Line 139 - ANNs are not unsupervised algorithms, but supervised in the most part;
14- Line 140 - ANNs are not derived from the human brain, but the nervous system of the superior organisms;
15- Line 140 - the basic units of an ANN ARE the neurons;
16- Line 145 - explain Equations 4 and 5. What are these variables? The same for Eqs. 6 to 11;
17- Eq. 9 - Asterisks means CONVOLUTION, not multiplication. Correct along the text;
18- Line 173 - it is almost impossible to understand the Equations because the authors do not explain the meaning
of each variable;
19- Figure 7 is wrong. The authors are using the term test wrongly. This is validation. Please consult
Haykin's book;
20- Which ANN was used? ANN is a class of algorithms, like MLP, RBF, ENS, ELM. The authors must define the
architecture used;
21- Line 201 - "best epoch value" - what dos it mean?
22- Line 211 - the authors are dealing with a forecasting problem, but using a classification metric to
evaluate the performance of the models considering just increase or decrease in the PM concentration. This
is not adequate;
23- How the authors separate the sets into training, validation and test?
24-How many times did the authors run the methods? It seams that they did this just once. The ANN models
must be executed at least 30 times because the dependence of the initialization;
25- The authors did not apply any statistical test to evaluate if the results are distinct;
26- There is no critical analysis regarding the results;
27- Why the authors used just 1 year of data?
28- the methodology is not well presented.
I believe the study has potential, but the manuscript present a lack of mandatory prerogatives for a
scientific study.
Author Response

(The authors gave the same response as above.)

Reviewer 3 Report
In this article, the authors developed a model that can predict whether the particular matter concentration will increase or decrease after a certain time for PM2.5 (fine PMs) generated by anthropogenic volatile organic compounds. The model developed in this study can be used to predict future PMs.
The authors made a remarkable research approach as there are social and environmental implications. The research is well organized and, a clear objective is set. Minor details on English grammar require review. Despite the solid scientific proposed methods and experiments, the authors do not highlight them in the best possible way.
The abstract is complete and well-structured. It explains the contents of the article clearly.
The introduction mentions the main contribution of the work. It provides sufficient up-to-date literature background.
In the Materials and Methods section, the authors describe, in detail, the models used and provide an adequate mathematical approach.
The authors miss the experiment setup. Please, provide the environment of the experiments.
The weakest part of the article is the analysis of the experiments. The authors have included too many figures without the required commentary. Discuss how your approach is better in terms of performance.
For discussion, the authors should clarify the limitations and the potential issues of this research.
The conclusion section could be further improved to highlight the simulation results, discuss future research directions and extensions of the study.
In summary, the article has good prospects and, the authors should emphasize the presentation of the results and their application in practice.
Author Response

(The authors gave the same response as above.)

Round 2
Reviewer 2 Report
The authors made a great improvement on the manuscript. I recommend acceptance.
Author Response
Title of the Paper: “A Particulate Matter Concentration Prediction Model Based on Long Short-Term Memory and Artificial Neural Network”
To the Editor and Reviewers:
We authors appreciate valuable comments made by the Editor and reviewers. Especially, Editor’s comment on English grammar and expression enhancement helped us to improve our submitted manuscript. We went through a scrutinized revision for correcting spelling and grammatical errors in the manuscript performed by a professional language editing expert as are specified in the editing list below. Again, thank you for your paying attention to this letter and we look forward to receiving your favourable final decision on the publication of our submitted manuscript.
Sincerely yours,
Seongju Chang, Ph.D.
Invited Professor
Department of Civil & Environmental Engineering, KAIST, Korea
T) +82-42-350-3627
F) +82-42-350-3610
M) +82-10-2757-4547
May 15, 2021
